# Contracting the private health sector in Thailand's Universal Health Coverage

Aniqa Islam Marshall [1,2☯]*, Woranan Witthayapipopsakul [1,2☯], Somtanuek Chotchoungchatchai [1], Waritta Wangbanjongkun [1], Viroj Tangcharoensathien [1]

1 International Health Policy Program, Ministry of Public Health, Nonthaburi, Thailand, 2 Faculty of Public Health and Policy, London School of Hygiene & Tropical Medicine (LSHTM), London, United Kingdom

☯ These authors contributed equally to this work.
* aniqa.marshall@lshtm.ac.uk

**Data Availability Statement:** All data has been provided as part of the submitted article.

## Abstract

Private sector plays an import role in health service provision, therefore the engagement of private health facilities is important for ensuring access to health services. In Thailand, two of the three public health insurance schemes, Universal Coverage Scheme and Social Health Insurance, contract with private health facilities to fill gaps of public providers for the provision of health services under Universal Health Coverage. The National Health Security Office (NHSO) and Social Security Office (SSO), which manage the schemes respectively, have designed their own contractual agreements for private facilities. We aim to understand the current situation of contracting private health facilities within UHC of the two purchasing agencies. This qualitative descriptive case study was conducted through document review and in-depth interviews with key informants to understand how they contract private primary care facilities, service types, duration of contract, standard and quality requirement and renewal and termination of contracts. Private providers make a small contribution to the service provision in Thailand as a whole but they are important actors in Bangkok. The current approaches used by two purchasers are not adequate in engaging private sector to fill the gap of public provision in urban cities. One important reason is that large private hospitals do not find public contracts financially attractive. NHSO classifies contracts into 3 categories: main contracting units, primary care units, and referral units; while SSO only contracts main contracting units. Both allows subcontracting by the main contractors. Contractual agreements are effective in ensuring mandatory infrastructure and quality standards. Both purchasers have established technical capacities to enforce quality monitoring and financial compliance although there remains room for improvement especially on identifying fraud and taking legal actions. Contracting private healthcare facilities can fill the gap of public healthcare facilities, especially in urban settings. Purchasers need to balance the right level of incentives and accountability measures to ensure access to quality of care. In contracting private-for-profit providers, strong regulatory enforcement and auditing capacities are necessary. Further studies may explore various aspects contracting outcomes including access, equity, quality and efficiency impacts.

**Funding:** This work was funded by the Asia Pacific Observatory on Health Systems and Policies (APO) grant through the University of Tokyo and the Thailand Science Research and Innovation (TSRI) under the Senior Research Scholar on Health Policy and System Research [Contract No. RTA6280007]. The APO had a role in approving the study design, data collection and analysis, and decision to publish. The views expressed in the submitted article are those of the authors and not their funders.

**Competing interests:** The authors have declared that no competing interests exist.

## 1. Background

Private sector's role in health has continued to grow and therefore plays an important role in the provision of health services in many countries [1, 2]. To achieve and advance Universal Health Coverage (UHC), engagement of the private sector in ensuring access to health services is crucial. An approach of purchasing organizations 'to find the best intervention to purchase, providers to purchase from and the best payment mechanisms and contracting arrangement to pay for such interventions' is a primary aspect of strategic purchasing [3]. Strategic purchasing involves organizational relationships between four key actors: government, purchasers, providers, and patients. Between purchasers and providers, the key policy objective is to improve provider performance for which a country will have its own set of governance structure shaped by its underlying values of the country's health and political systems [4]. Despite its importance, strategic purchasing function has been neglected in pursuing UHC in low- and middle-income countries [5].

Thai Universal Health Coverage (UHC) is constructed from a publicly financed health system, consisting of three main public insurance schemes that offer full-service coverage. The three schemes are the Civil Servants Medical Benefits Scheme (CSMBS) for civil servants and their dependents managed by the Comptroller-General's Department (CGD), Social Health Insurance (SHI) for private sector employees managed by the Social Security Office (SSO) and the Universal Coverage Scheme (UCS) which covers 70.8% of the population in the country managed by the National Health Security Office (NHSO) [6].

Thailand's health service delivery system is publicly dominant as a result of early and sustained public health system development since 1970s. The public health infrastructure was bolstered through the complete geographical coverage of health centres in all sub-districts, secondary care hospitals in all districts, tertiary care facilities in all provinces and more advanced referral hospitals in all 13 public health regions [7]. This hierarchical public service delivery system provided a promising platform for successful UHC implementation, with primary care units (defined as district health system contractor networks) acting as gatekeepers for higher level care [7, 8].

However, the district health service delivery system does not exist in major cities, giving room for private clinics and hospitals to fill the primary care gap needed to deliver health services under UHC. Although there are only 382 private hospitals compared to the 2,054 public hospitals in the country, private health facilities contribute 72% of all health facilities [9, 10]. The large majority of these facilities are private clinics which play a significant role in the provision of primary health care under UHC (See Table 1). In Bangkok, there are only 69 primary care centres owned by the Bangkok Metropolitan Authority, in contrast to the 6,296 private clinics providing primary care services in the city.

While all public facilities are required to serve as providers for the public insurance schemes, UCS and SHI also engage private providers to serve their urban members where there are gaps in public provider. Due to different pace of development, SSO in 1991 and NHSO in 2001, have independently designed their own contractual agreements for private facilities that wish to provide services for their beneficiaries. The CGD directly disburses public healthcare facilities for primary care services; and reimburses inpatient care through Diagnostic Related Group system; but does not permit reimbursement of primary care services through private providers.

Although NHSO and SSO contract public and private providers, this study focuses on private contracting—one of the neglected research areas. To better understand the current situation of contracting private health facilities within UHC and identify gaps and challenges, this study compares the two purchasing agencies, namely NHSO and SSO on how they contract

**Table 1. Numbers of health facilities, hospital beds and medical doctors in the public and private sectors categorized by facility types and geographic regions.**

|  | Public | Private |
|---|---|---|
| Facilities |  |  |
| • Total | 11,959 | 31,286 |
| Hospitals | 2,054 | 382 |
| <90 beds | 814 | 170 |
| 91–500 beds | 221 | 158 |
| 501–1000 beds | 46 | 0 |
| >1000 beds | 11 | 0 |
| No data on number of beds | 0 | 54 |
| Sub-district Health Centres | 9,905 | 0 |
| Clinics | 961 | 30,904 |
| • Bangkok | 261 | 6402 |
| Hospitals | 193 | 106 |
| Bangkok Metropolitan health centres | 68 | 0 |
| Clinics | 0 | 6296 |
| Hospital beds |  |  |
| • Total | 127,275 | 32,566 |
| • Bangkok | 17,051 | 12,594 |
| Medical doctors |  |  |
| • Total | 32,428 | 31,767 |
| • Bangkok | 6,496 | 16,508 |

Source: Ministry of Public Health [9, 10]

private primary care facilities, service types, duration of contract, standard and quality requirement and renewal and termination of contracts.

## 2. Methods

### 2.1 Scope of study

This study covered UCS and SHI which have contractual arrangements with private primary healthcare providers.

### 2.2 Study design

A qualitative descriptive case study was applied through document review and in-depth interviews with key informants. Data was extracted and analysed into themes based on an adapted conceptual framework for evaluating contracting-out health services [11].

Ethics approval was obtained from the Institute for the Development of Human Research Protections of Thailand; the certificate number IHRP2020012 on 28 January 2020. Written informed consent was obtained before commencing each interview and key informant information was de-identified to maintain anonymity.

### 2.3 Context and setting

Given the private sectors unique role in the city, Greater Bangkok Metropolitan Area was selected as the study setting [12]. Almost half the private facilities in the country are geographically concentrated in the central region, of which 43% are in Bangkok, where all types of

private facilities including clinics make up 96% of all health facilities in the city (See Table 1). Though most private facilities are clinics or smaller less-than-90-bed hospitals and only 20% of hospital beds in the country are in private facilities, in Bangkok, approximately 42% of all hospital beds are found in private hospitals. Additionally, while the number of medical doctors between the public and private facilities throughout the country is similar, in Bangkok, over 70% of medical doctors work in private facilities.

## 2.4 Data collection

**2.4.1 Semi-structured in-depth interviews.** Semi-structured in-depth interviews with key informants were conducted to explore the contractual agreements between public purchasing agencies and private providers as well as their experiences and perceptions. Interview guides were separately developed for each of the two stakeholder groups:

First, for purchasing agencies, the major areas of enquiry included: a) profile and characteristics of contracted private facilities; b) contractual requirements, terms, conditions, and regulations; c) effectiveness and challenges in engaging private providers. Stakeholders were recruited through direct request for interviews to the insurance scheme offices. Senior management officers with knowledge on and involvement in implementing contracts with private sector facilities were interviewed.

Second, for key informants from private providers, the major areas of enquiry included: a) general profile and capacity of the health facility; b) requirements, conditions and standards to join public insurance; c) motivations and challenges in contracting with public insurance. Health facilities in the Bangkok Metropolitan Area that have ever undergone contractual agreements with NHSO and/or SSO were contacted from lists of registered private facilities in the UCS and SHI schemes for in-depth interviews [13, 14]. Senior management executives and owners of private health facilities with experience in undergoing contractual agreements with NHSO or SSO were interviewed.

Verbal and written consent was obtained from all participants. All interviews were conducted in Thai. Each interview was audio-recorded and detailed field notes were documented.

**2.4.2 Document review.** Documents published between 2002–2021, including policies, reports, statistics, presentations and publications on private health facilities contracted within the UHC schemes were searched through relevant government agency websites, including the Royal Thai Government Gazette, NHSO, SSO and National Statistical Office (See Table 2). Documents were selected based on the title's relevance to the research topic. Additionally, documents relevant to key informant discussions (such as unpublished reports, regulations, forms) were requested and obtained directly from key informants. Selected documents were reviewed in full and included if any data applicable to the data analysis themes were identified.

**2.4.3 Data analysis.** We adapted conceptual framework for evaluating contracting-out of health services by Liu et al. [11]. Engagement of private sector is analyzed by the characteristics of the purchaser organizations, the contractor providers, and the contractual relationship, which includes six components: 1) types of services; 2) contract formality; 3) contract duration; 4) provider selection; 5) payment mechanism and 6) performance and quality assurance (See Fig 1).

Audio recordings from each interview were transcribed. Data from the transcriptions and the field notes were summarized then manually coded and categorized based on relevance to each analysis component. Data obtained from documents were also categorized into each component. Information provided by key informants were verified and triangulated with review of relevant documents. We gave priority to documents over verbal information especially on factual evidence. When supporting documents were available, interview findings

**Table 2. Documents relevant to contracting private health facilities in UHC.**

| Type | Document number and title | Organization |
|---|---|---|
| Report | 1. Consideration for approval as a service provider within the National Health Security System [15] | National Health Security Office |
| | 2. National Health Security Office annual report [16] | National Health Security Office |
| | 3. The management of provider payments in Universal Coverage Scheme in Thailand [17] | National Health Security Office |
| | 4. Thailand UHC & overview of the Universal Coverage Scheme of the National Health Security Office [6] | National Health Security Office |
| | 5. Fund Management Manual: National Health Insurance [18] | National Health Security Office |
| | 6. Audit system for the Universal Coverage Scheme in Thailand [19] | National Health Security Office |
| | 7. Local area and list of hospitals and clinics [14, 20] | Social Security Office |
| | 8. Annual report of Social Security Office [21] | Social Security Office |
| | 9. Recruitment of private sanatoriums providing medical services to insurers [22] | Social Security Office |
| Policy | 10. Social Security Act [23] | Social Security Office |
| | 11. National Health Security Act [24] | National Health Security Office |
| | 12. Regulations on the rules, procedures and conditions for registration as a service unit and service unit network [25] | National Health Security Office |
| | 13. Notification of the Social Security Office: standardization of sanatoriums providing medical services to insurers [26] | Social Security Office |
| | 14. Notification of the Ministry of Public Health: characteristics of medical facilities and standards [27] | Ministry of Public Health |
| Database | 15. Geographic information system on health [9] | Ministry of Public Health |
| | 16. Registered private hospitals [10] | Ministry of Public Health |
| | 17. Universal Coverage Scheme information [13] | National Health Security Office |
| | 18. Health and Welfare Survey [12] | National Statistical Office |
| Relevant material | 19. The social security scheme in Thailand: what lessons can be drawn? [28] | Tangcharoensathien et al. |
| | 20. Developing Health Care Quality Indicators and improving the QOF Program for the Thai Universal Health Coverage [29] | Roongnapa et al. Health Intervention and Technology Assessment Program |
| | 21. Thailand Healthcare Accreditation: a journey [30] | Supachutikul, Aunwat. Healthcare Accreditation Institute |
| | 22. Comparison of alternative relative weights for diagnosis-related groups [31] | Cotterill et al. |
| | 23. Defining the benefit package of Thailand Universal Coverage Scheme: from pragmatism to sophistication [32] | Tangcharoensathien et al. |
| | 24. Patient-level costing for the Thai Diagnosis Related Group in Thailand: a micro-costing approach [33] | Khiaocharoen et al. |
| | 25. Achieving universal health coverage goals in Thailand: the vital role of strategic purchasing [34] | Tangcharoensathien et al. |
| | 26. DRG coding practice: a nationwide hospital survey in Thailand [35] | Pongpirul et al. |

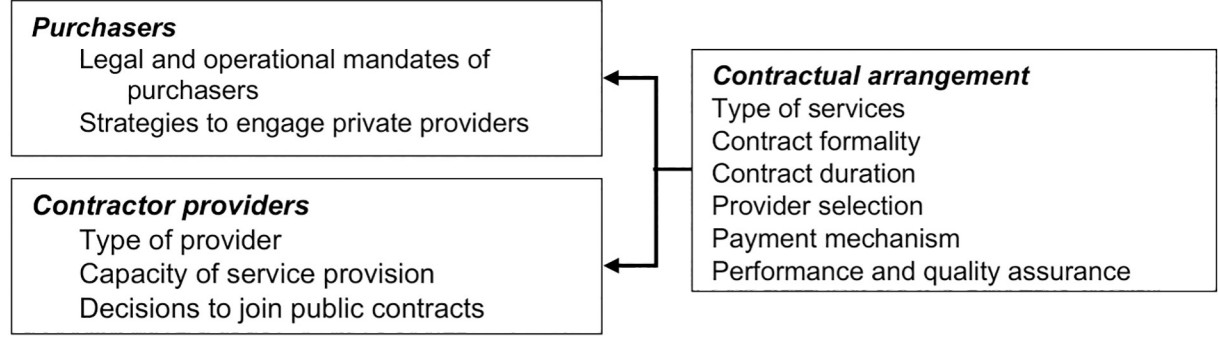

**Fig 1. Conceptual framework for analysis.**

were used to support our understanding about policy intentions, underlying problems, procedures and steps, time sequences, or example cases. The interview was, however, the only source for the findings about perceptions of purchasers on challenges in contracting private providers, and providers' motivations and challenges in contracting with NHSO and SSO. Narrative analysis was applied. Minor quantitative analysis was also conducted by reporting values, ratios and percentages of statistical data obtained from the document review.

## 3. Results

A total of seven key informants were interviewed between Aug 2019-Mar 2020. Table 3 shows their profile. Twenty six documents were included in the analysis (Table 2).

We organized our findings from interviews and document review into four sections: 1) nature of two purchaser organizations; 2) contractor provider categories; 3) contractual arrangement; and 4) purchaser and provider perspectives on effectiveness of the current private provider engagement strategies.

### 3.1 Nature of two purchaser organizations

Two purchaser agencies, the SSO and the NHSO, are involved in contracting with private health facilities to fill gaps in service provision notably in urban areas in addition to mandatory services by public providers.

The SSO, a department in the Ministry of Labour, is a governmental agency, mandated by the 1990 Social Security Act. A tripartite Social Security Committee, consisting of seven government *ex-officio*, seven representatives from employers and seven representatives from employees, is the governing board of Social Security Scheme [23]. The Board is supported by a Medical Committee as advisory role on matters related to Social Health Insurance [28].

The SSO has several legal mandates. It collects monthly payroll-tax contribution via wire transfer by employers and employees and annual contribution from the government budget allocation, as a tripartite contribution. The Office is required to register all private employers and employees nationwide, maintain database on employee's salary, as a basis for estimating contributions and provide payments such as for sickness or pension benefits for those retired [23].

Operating out of provincial social security offices throughout the country and 12 branch offices in Bangkok, SSO contract public and private health facilities on a competitive basis. In addition to medical benefits, the SSO also manages unemployment, child allowance, disability, death compensations and old-age pension as part of a comprehensive social security system.

The NHSO, a public organization established under the 2002 National Health Security Act, is mandated to implement the UCS. The National Health Security Board (NHSB) is the governing body of UCS. Out of the total 30 Board members, five are citizen representatives

**Table 3. Profiles of key informants.**

| Stakeholder Group | Number and Position | Gender |
|---|---|---|
| Purchasing Agency | 1 Senior Management Officer, NHSO | Female |
| | 1 Senior Management Officer, Regional NHSO | Male |
| | 1 Senior Management Officer, SSO | Female |
| Private Provider | 1 Hospital owners | Female |
| | 1 Hospital management executive | Male |
| | 2 Clinic owners | 2 Males |

selected among themselves from the nine civil society organization constituencies registered with the Ministry of Interior [24].

The NHSO administers the fully tax-financed National Health Security Fund, through an annual budget negotiation at the Finance and Budget Sub-committee of the NHSB where Budget Bureau is one of the members [6]. In addition to the NHSO headquarters, there are 13 regional offices where private sector providers can apply to be registered into the National Health Security System (NHSS) to provide health services to UCS members.

## 3.2 Contractor provider categories

In 2021, the NHSO revised the classification of health facilities that are eligible to join the NHSS into three categories [25]:

- Main Contracting Units which enter into contractual agreement with and are account to NHSO. It has the capacity to provide comprehensive prevention, health promotion and curative services and shall have a referral hospital backup.

- Primary Care Units which may or may not enter into direct contractual agreement with NHSO. It forms into network of primary care units and affiliate with a main contractor unit. It provides services to the catchment population and is paid by NHSO or the main contracting unit.

- Referral Units who provide treatment for referral cases from the main contracting units and paid by them.

In the past Joint Service Providers were contracted to provide specific services such as dental, rehabilitation and pharmacy services, etc. However, as the interpretation of the revised legal text did not allow for providers to be categorized as "joint service providers", they are labeled as referral providers in practice.

In 2019, there were a total of 12,337 public and private health facilities registered with NHSO nationwide but less than 5% (580/12,337) were private (of which 301, 281 and 266 were Primary Care Units, Main Contracting Unit and Referrals Units). However, in Bangkok, private facilities made up nearly 70% of all facilities providing health services to UCS members. Of the 274 private providers, 193 facilities serve as both primary care units and main contracting units which entered into a contractual relationship with NHSO, 78 facilities accepted referrals from main contracting units, see Table 4 [13].

**Table 4. Facilities contracted within the universal health coverage system.**

| | | NHSO Facilities | | | | SSO Facilities |
|---|---|---|---|---|---|---|
| | Facilities | Primary Care Units | Main Contracting Unit | Referrals Units | Total Units Registered* | Main Contracting Units (hospitals) |
| Thailand | Public | 11,449 | 1,079 | 1,074 | 11,757 (95%) | 163 (67%) |
| | Private | 301 | 281 | 266 | 580 (5%) | 79 (33%) |
| | Total | 11,750 | 1,360 | 1349 | 12,337 (100%) | 242 (100%) |
| Bangkok | Public | 95 | 90 | 43 | 134 (33%) | 19 (40%) |
| | Private | 193 | 193 | 78 | 274 (67%) | 28 (60%) |
| | Total | 288 | 283 | 121 | 408 (100%) | 47 (100%) |

Source: National Health Security Office [13] and Social Security Office [20].

Retrieved 28th January 2020

*Note: Some facilities are registered by NHSO for more than one contract type. Therefore, the total unit registered exceeds the sum of three types of contractors.

All public and private hospitals with contractual agreements with SSO are called main contracting units; there is no other differentiation. These main contractors are hospitals that satisfy minimum resource and standard requirements set by SSO (such as facilities having at least 100 beds and 12 clinical specialties) and receive direct payments from SSO [26].

Applications from private providers to provide health services for SHI members, are assessed at headquarter and provincial branch offices. In 2020, SSO contracted 242 hospitals in the country, with 33% being private. Of all private facilities, 35% are located in Bangkok, where private health facilities constitute almost 60% of all health facilities. Approximately 40% of SHI members in the country and 52% in Bangkok choose to register with a private main contractor hospitals [20]. (Table 4) Although a total of 242 providers for SHI appeared small, this number only represented main contractors, and did not include subcontractors and supra-contractors which also provided health services to SHI members.

SSO allows main contractors to enter into agreements with 'supra-contractors' and 'sub-contractors' which serve in their service network. Supra-contractors are referral hospitals or specialized hospitals that agree to provide services to patients referred from the main contractors when patient needs are beyond their capacity; these are usually regional hospitals, teaching hospitals, or specialized hospitals in the same or neighbouring provinces. Sub-contractors are facilities that support main contractors in providing primary care services, provided service standard are maintained, which can be hospitals or clinics located in the same geographical areas. SSO does not directly contract sub- and supra-contractors, and only the main contractor hospitals are legally accountable to the SSO. Payment of services to sub- and supra-contractors are internal process by main contractor hospitals.

## 3.3 Contractual arrangement

**3.3.1 Types of services.** UCS members are entitled to a comprehensive benefits package including disease prevention, diagnosis, medical treatment, health promotion, rehabilitation and dental services, as well as Thai traditional and alternative medicine. In addition to basic inpatient and outpatient services, the NHSO's targeted and special services include thrombolytic therapy for ischemic stroke patients, cataract lens replacement surgery, transplantations (heart, liver, stem-cell), knee arthroplasty, thalassemia, HIV/AIDS including Anti-retroviral treatment, renal replacement therapy, diabetes and hypertension, chronic psychiatric care and long-term care for elderly [16, 32]. There are no copayments for UCS members as NHSO fully funds all services.

Similarly, the SSO covers all expenses for medical examination and diagnosis, medical treatment, admission to and treatment in medical establishments, medicine and medical supplies, and ambulatory services. Coverage includes organ transplants, HIV/AIDS, haemodialysis, dental treatment, transplantation (bone marrow, kidney and cornea), and renal failure treatment. SSO contracts with the main contractor hospitals to provide inclusive outpatient and inpatient services using an annual capitation fee plus some additional payments for outpatient and inpatients (see detail in sub-section 3.4). Therefore, contracted hospitals must meet all required capacities.

**3.3.2 Contract formality and duration.** NHSO renews annual contractual agreements with private main contracting units, on a condition that they maintain NHSO standards and pass annual inspections. Call for new health facility applications are announced annually, with applications assessed at NHSO's regional offices. NHSO can undergo contractual agreements with all three categories of service providers (as described in section 2), with direct payment mechanisms established for the contracted facilities. A health provider may apply to register as one or more facility type [15]. This has been the case for facilities in Bangkok where most

private contractors located. However, NHSO may not make a direct contract with primary care units and leave primary care provision in responsibilities of the Main Contracting Units. NHSO applies this practice to most health centers belong to Ministry of Public Health (MOPH) in all 76 provinces outside Bangkok. It leaves MOPH's district hospitals to manage service and financial agreements with their own subsidiary health centers. Contractual documents with private providers are legally binding with financial penalties for non-adherence, while agreements with public providers do not have provisions on penalties.

Contractual agreements between the SSO and private main contracting units are legally binding contracts which are renewed annually with new rates, terms and conditions if applicable. Unlike agreements with public main contracting units where contracts are automatically renewed, private providers must undergo annual re-application, review and re-registration process [26]. Calls for new applications and renewals of the incumbent contractors are announced between April and May through the SSO's website and invitation letters to previously contracted providers. Interested providers submit their applications for review to SSO headquarter or its provincial branch offices. Private providers that are selected must sign contracts by October to provide services from January to December of the upcoming year [22]. For supra-contractors and/or sub-contractors, SSO is not involved in the agreement process; it is left to internal management of the main contracting units. However, main contractors are required to notify SSO upon application with details about their sub- and supra-contractors' facility licenses, service scope, and their agreement and consent documents.

New interventions or medicines are introduced into the benefit package and inflation from labour and medical supplies cost can affect the cost of service provision and capitation rate. Both NHSO and SSO apply annual contracts with new rate of capitation and other renewal conditions [17].

**3.3.3 Selecting providers for contracting.** A minimum eligibility criterion for a main contracting unit has been established by the NHSO. This includes a) being a health service provider that is already operational, b) must have received a unique health service provider identification number generated by the Bureau of Policy and Strategy, Ministry of Public Health, c) if the health service provider is a primary care unit, it must have an agreement with a host service provider that serves as its main contracting unit, d) if the health service provider is a main contracting unit that is not also registering as a referral unit, it must have a referral unit already registered within the National Health Security System. All NHSO regulations aim for a seamless referral of patients who require secondary or tertiary hospital services [15].

Additionally, main contracting units must have a doctor to population ratio of 1:10,000, referral units must have a minimum of 30 beds and have at least four main specialties, namely obstetric and gynaecology, general surgery, internal medicine and paediatric, while private sector providers must be licensed and re-licensed annually by the Bureau of Sanatorium and Art of Healing, Ministry of Public Health [15].

Prior to entry into contractual agreement, the NHSO's service inspection unit assesses the applicants registering as main contracting units and/or referral units based on a nationally set standards and guidelines developed jointly by NHSO, Ministry of Public Health and relevant stakeholders. Criteria of pre-entry assessment differs by types of contracting units [15]. NHSO also re-assess the contractors annually to ensure these standards are maintained. The one-day assessments are conducted by trained public sector medical doctors and nurses for both new registrations and annual re-assessment inspections. In practice, the main contracting units are inspected and assessed by staff from their affiliated referral hospitals. Facilities that do not pass either the initial inspection or annual inspection are requested to make either general or specific urgent improvements within 7 days otherwise they are removed from the NHSS.

The private main contractors for Social Health Insurance must meet the following requirements: a) having at least 100-bed capacity; b) ability to arrange referrals to ensure patients are treated to the utmost medical care they need; and c) having at least 12 medical specialties [Internal medicines, General Surgery, Ob-Gyn, Paediatrics, Orthopaedics, Preventive Medicines, ophthalmology, anaesthesiology, Ear Nose and Throat, Radiology, Rehabilitation, Emergency medicine or neuro-surgery] [26]. The facilities that do not meet all criteria may be contracted on an exceptional circumstance as advised by the SSO's medical committee; for example, if it is the only health facility in the underserved areas in particular public contractor. SSO also sets standards for specific areas such as general facility standards, emergency services, outpatient / inpatient / intensive care, surgery, pharmacy, and medical records [26]. For the registration of new applicants, main contractors must also be an accredited hospital either through the Joint Commission International (JCI) or Thailand Healthcare Accreditation Institute (HAI). These requirements and regulations are not enforced on sub-contractors and supra-contractors which are separately contracted by main contracting facilities.

**3.3.4 Provider payment mechanisms.** Both NHSO and SSO make direct payments to facilities that have undergone direct contractual agreements and implements a blended payment model.

From the NHSO, Main contracting units providing outpatient services to UCS members are paid prospectively one month in advance as well as general health promotion and disease prevention services. The capitation payment is calculated for each facility by multiplying an age-adjusted differential capitation rate with the number of members registered to the main contracting unit; the more number of elderly and children under five years old receives slightly higher outpatient capitation rate. The total budget for outpatient is calculated for each contractor unit based on the population in the catchment area registered with each contractor. Age adjustment is applied to 80% of capitation rate while the remaining 20% is flat rate equally for any main contracting unit [17].

In-patient services for UCS members are paid through retrospective claim and reimbursement by applying Diagnosis Related Groups (DRGs) System under a global budget. Reimbursement payments are also used for special health promotion and disease prevention services in cash and in kind for specific services such as vaccines, anti-retroviral drugs and peritoneal dialysis solutions in the form of medicines and medical supplies [17]. For example, NHSO authorizes Government Pharmaceutical Organization to purchase peritoneal dialysis solution; contracts Thai Post Distribution to deliver these solutions to patients' home nationwide; pays Main Contracting Units where patients registered with for supervising and supporting patients at home.

Providers are responsible for filing their reimbursement claim forms within 30 days after the inpatient was discharge from hospital, through the NHSO e-Claim program managed by the NHSO Bureau of Information Technology. The claim is then verified by the Bureau of Fund Allocation. Upon satisfactory verification, providers are compensated by the Bureau of Fund Accounting through electronic bank transfer directly to the health facility [17].

In addition to capitation for outpatient and DRG reimbursement for inpatient, a fee schedule under a global budget is applied for high-cost services that may pose a financial risk to health facilities, such as knee prosthesis and high-cost medication such as recombinant tissue plasminogen activator for the treatment of acute ischemic stroke [17].

NHSO also applies a fixed fee per patient for some non-communicable diseases screening services, and a point system with global budget for acute diseases, emergency services, Thai traditional medicines and rehabilitation services. Activity-based contracts for area-based and community-based health promotion and disease prevention services to meet the needs of local health problems are managed by regional NHSO offices and paid as project-based "block-

grants". The grant is allocated to sub-district health centres [17]. All of these blended payment to providers aims to improve access to high cost but effective interventions and minimize the weakness of potential "under provision of services" of the main mode of capitation payment.

The SSO applies a mix of payment methods, which include capitation, point-system, DRG, and fee-schedule. The DRG relative weight is the relative resource intensity of each inpatient. It is the ratio between average resource for one DRG and total resource for all DRG [31]. Each disease group was assigned with a relative weight; the higher the DRG weight, the more resources are required to treat a patient, meaning the more payment to the providers. Thailand had applied DRG systems for more than two decades [33]. Outpatient and inpatient services are paid monthly based on an annual fixed capitation rate per SHI member registered to the facility. Additional per-capita payments are also added for chronic illness patients. For inpatient care with an adjusted relative weight over 2 under the DRG systems, services for these patients are paid by applying DRG with global budget. For inpatient with relative weight below 2, it is covered by the capitation rate. Supra- and sub-contractors are paid directly from main contractor units, not SSO, with rates and terms mutually agreed upon between the main contractor and the supra- and/or the sub-contractors.

To incentivize providers to provide adequate treatment for out-patients with chronic diseases, an additional 453 THB per registered SHI member for each main contractor hospital was offered. Half of this amount is paid on a monthly basis for 11 months calculated from the number of points per capita member earned by providing outpatient services for the 26 chronic diseases by each contractor. On the last payment in month 12th, the contractor hospitals receive the remaining half of 453 Baht based on its points per capita earned as a share of total national points per capita by all contractors in a year.

Further, certain equipment such as orthopedic prosthesis can be reimbursed using a fee-schedule. SSO also pays an additional 80% of the amount of expense over 1,000,000 THB for high-cost in-patient service of more than a million Bath. This complex mix of additional payment method applied by SSO aims to incentivize adequate treatment for more complex and high-cost outpatient and inpatients.

**3.3.5 Performance requirements and quality of care.** The NHSO has introduced various mechanisms to monitor health facility performance and ensured service quality. The Health Service Standard and Quality Control Board (HSQCB), one of the two governing bodies of the NHSO, sets and monitors the standards and quality of health providers. The Quality Control subcommittee of the HSQCB sets both national and regional standards to ensure applicability to different regional contexts, including criteria for new entry and contract renewal [6]. Health service providers that do not meet the assessment criteria are unable to enter to contract, while facilities that fail annual inspections are subject to dismissal or contracts not renewed. Apart from the new entry and renewal requirements, NHSO also conducts unannounced 'surprise visits' to health facilities, usually for facilities suspected to be non-compliant to NHSO requirements based on document audits, complaints received from UCS members, and random checks with users [19].

In 2014, to increase quality of services, the Quality and Outcomes Framework (QOF) for primary care providers links payments with their performance. The Thai QOF, which aims to reflect the quality of health promotion and disease prevention, primary care, administrative services, and performance of services, consists of six national health service performance indicators set by the NHSO in collaboration with the Ministry of Public Health and Thai Health Promotion Foundation and not more than five regional specific indicators approved by regional committee based on health and quality challenges faced by the region [29]. See Box 1.

In 2004, The Healthcare Accreditation Institute (HAI) of Thailand initiated a new stepwise standardization process to recognize hospitals of varying quality standards [30]. Step one

Box 1. Quality outcome framework indicators, fiscal year 2021 [18]

A. National Core Set of Indicator

1. Percent of population age 35–74 years are screened for diabetes using fasting blood sugar

2. Percent of population age 35–74 years are screened of hypertension

3. Percent of pregnant women having antenatal care before 12 weeks gestation

4. Cumulative percent of women 30–60 years old having cervical cancer screening in the last five years

5. Percent of outpatients covered by rational use of antibiotics for a) acute diarrhea, and b) acute respiratory infection

6. Hospitalization rate of five ambulatory care sensitive conditions (epilepsy, COPD, asthma, diabetes and hypertension)

B. Samples of Regional Specific Set of Indicators

1. Percent of children 9, 18, 30 and 42 months old received developmental screening, and percent children detected with delayed developed received treatment within 30 days

2. Percent of home-bound and bed-ridden elderly received home visits

3. Percent of depressive patients have access to psychiatric treatment

4. Percent primary school children who are obese,

5. Percent primary school children received oral health screening

6. Percent of adults 18–59 years who have normal body mass index.

accreditation was granted for facilities which established preventative measures and risk identification systems in place; step two for facilities meeting key Hospital Accreditation standards of quality assurance and quality improvement; and step 3 was granted to hospitals meeting all Hospital Accreditation standards.

In fiscal year 2007, NHSO earmarked a budget of 0.76 Baht per capita member (35.7 million Baht in total) as additional incentives based on the achievement of different accreditation status. NHSO applied a scoring system with a maximum of score of 5 for hospitals with full accreditation, score of 2 for those achieving step 2 and score of 1 for those achieving step 1 from the HAI accreditation. Fully accredited primary health care facilities have been entitled to receive a grant of up to 30 Baht per capita multiplied by catchment population it served [34].

In contrast to the NHSO, the SSO does not proactively incentivise hospitals for continuous quality improvement. However, the Office relies on external evaluations and certifications from the Healthcare Accreditation Institute or Joint Commission International (JCI) as conditions for private hospitals to enter into agreements with SSO. Additionally, SSO conducts assessments and quality control audits, with main contracting units assessed annually.

**3.3.6 Enforcement of contract through audit.**   NHSO pioneered three categories of audit. First, coding audit is used to detect upcoding of DRG relative weight, DRG creep, which is a deliberative process to gain higher reimbursement [35]. DRG claims, randomly selected or targeted samples of outliers, were retrospectively reviewed by trained independent auditors. Overclaimed amounts were asked to return to NHSO otherwise legal actions would be taken in several cases. Second, billing audit identifies outstanding or abnormal claims by health facilities for services paid outside capitation or DRG such as Pap Smear for cervical cancer. By 2021, total 2272 auditors were trained and actively function. Though SSO does not develop auditing system, it applies NHSO coding audit though not systematic and regularly.

SSO has never terminated contracts though poor performing facilities voluntarily leave the Scheme. Prior to 2019, NHSO has terminated a few private contractors in Bangkok due to fraud and failed quality standard. In the past, NHSO only pressured the poor performance not to renew the contract.

## 3.4 Effectiveness of the current private engagement

Interview findings uncovered various issues on effectiveness of the current strategies that NHSO and SSO use to engage private providers. We separated results into purchaser and provider perspectives.

**3.4.1 Purchaser perspective.**   Overall, the two purchasers felt that the current levels of private provider participation were still inadequate to fill service gap. Although several forms of financial and non-financial incentives were in place to attract private providers, these incentives were not seen attractive to all. First, financial payment rates were generally standardized i.e., applied to all facilities of the same type. Whether these incentives were attractive enough also depended on provider factors. Capitation payment, the major source of hospital revenue, offered predictable and steady flow of incomes. However, a significant drawback of capitation method was inadequate risk sharing among smaller hospitals or those recently operated which were unable to register the maximum or nearly maximum number of populations for their capacities. In such circumstance, small pools of UCS member took higher financial risk when they got some serious chronic illness cases and were unable to survive. Second, smaller hospitals operate with higher cost than larger hospitals or hospitals formed into chains who negotiate large-scale medicines and supplies purchase with lower cost. Third, new contractors usually experienced difficulties competing with long-standing reputable contractors for registered patients. Finally, some private hospitals joined public contracts to secure funding and gain reputation and, after some years, left for more profitable markets.

In addition to insufficient number of participating providers, purchasers also viewed that the current accountability mechanisms still provided some opportunities to exercise profit-making behaviors such as cherry picking especially among hospitals registered in the stock market. A measure to control quality beyond standard requirement was also lacking. For example, the NHSO regulations only allowed contract termination based on two conditions i.e., when the contractors requested to leave or when they failed the standard requirement. However, in the situation that the purchasers still wished for greater participation, more stringent criteria or lower autonomy would counteract the expected target.

**3.4.2 Private sector provider perspective.** Providers raised several push and pull factors underlying decisions to join public contracts. Representatives from small hospitals and private clinics agreed that they could generate sufficient-to-survive revenues from public contract. These providers also satisfied with different revenue generating activities apart from capitation methods. These included fixed fee for services such as individual disease prevention activities, home visits, mobile services as well as financial rewards from achieving QOF and higher quality standard. A few also mentioned intrinsic values namely fulfilling their altruism and social responsibility to serve people in local communities, opportunities to work with intersectoral parties, and social recognitions. Market competition was also involved when neighboring providers participated in public contract pushes them to join the race. However, meeting NHSO standard requirements such as minimum number of required health professionals, computer systems, being subjected to audit was seen as a barrier especially for physician-owned private clinics.

On the contrary, a representative from a large hospital chain shared reasons for not joining UCS or SHI provider network. By and large is the feeling that private providers were treated unfairly by purchasers. For instance, they were not able to negotiate payment rates. The global budget system made DRG relative weight unpredictable and revenue predictability was vital to run private businesses. They also felt that public providers should share registered population to private providers proportionately to infrastructure size. It was also (mis) perceived that public providers received staff salary on top of the same payment rates. Lastly, regulations were unequally enforced between public and private facilities. Despite several disincentives, some private providers had to join public contract because their locations were among SHI users such as locating near factories.

## 4. Discussion

Our findings suggest that private providers make up a small contribution to the service provision under public insurance system as a whole but they are important players in Bangkok. The current approaches that NHSO and SSO apply still have room for improvement in order to adequately engaging private sector to fill the gap of public provision under UHC.

The nature of private-dominated healthcare market in Bangkok gives NHSO and SSO limited choice to contracting the existing private sector, mostly for-profit private clinics and hospitals. The capitation rate and DRG payment by NHSO and SSO reflect the full cost of operation including capital deprecation. These rates, terms and conditions are applied equally to public and private contracting providers. High-end private hospital chains listed on the stock exchange, certified by Joint Commission International or HAI are able to generate larger and increased trend of revenue from medical tourists do not join public contracts. These hospitals have larger capital through mergers or acquisition and competitive bulk purchasing gains technical efficiency. Many small to medium size private facilities, not able to compete with large hospitals, are contractor providers for NHSO and SSO as they find their contract terms attractive. Secure and steady flow of fund supports their annual business plan. However, some small private contractors especially clinics may have higher operating cost than revenue from contracting, partly because they have limited capacity to negotiate price through bulk purchasing.

A realist review on strategic purchasing identifies three factors influence the performance of contractor providers: provider accountability mechanisms; provider autonomy; and power balance between purchasers and providers [4]. Our study found similar evidence as in the literature. Regarding the accountability mechanisms, NHSO and SSO used both contractual and extracontractual mechanisms to steer providers to its expected targets. The contracts of the two purchasers are similar in terms of application and renewal process, contract duration,

quality prerequisites and inspection. NHSO differentiates three distinct contractor provider types while SSO applies a simpler contractual arrangement with the Main Contracting Hospital only. These contracts are effective in terms of ensuring basic capacities to provide particular level of care or care specialty and standard quality. However, both purchasers allow subcontracting to other facilities outside the purchaser supervision and standard control. This can lead to suboptimal quality of service which is difficult to monitor in the absence of information sharing channels. It also prevents the purchasers from monitoring fraud and unethical profit-making behaviors among subcontractors.

The mixed methods of provider payments appeared to be adequate in ensuring responsiveness to population health needs. The use of various payments in addition to basic capitation rate and DRG to incentivize adequate level of service, for example, additional per capita payment for 26 chronic illness patients, DRG payment for inpatient care with relative weight greater than 2.0, and fixed fee payment for high cost interventions. NHSO also boost quality and standard through differential bonus for contractor providers with higher accreditation status.

Audit is an important tool to ensure accountability and transparency. Prior to 2019 at the time of the interview, NHSO only terminated a few private clinics in Bangkok due to fraud and failed quality. However, in 2020, NHSO uncovered a large scandal by 288 private clinics in Bangkok, dental clinics and hospitals; incurring a total loss of 691 million Baht (US$ 23 million) to NHSO from billing audits [36] False billing was manipulated by the use of "ghost patients", i.e., using citizen ID number of others who were not users of services to claim for special payment such as screening of diabetes, hypertension and provision of dental services. The current NHSO's authentication of users using the unique citizen ID number was not very effective to filter frauds if the ID number used was also belong to UCS members and also time consuming. NHSO filed these cases to the criminal court, terminated contracts, banned the facilities from future contracts, as well as reviewed and strengthened the terms and conditions for contractual agreements and improved auditing systems [25]. In the future, the use of biometrics such as finger prints or retina scan can be more promising.

Prior literature highlighted the importance of organizational capacities of purchasers to implement strategic purchasing [5]. This is particularly crucial for effective accountability mechanisms as they require mutual agreements and necessary actions from both purchaser and provider sides. It needs, for instance, timely submission of electronic claims and all essential information from the provider contractors and timely payment transfers from the purchasers to enforce these accountability mechanisms. Based on our evidence, both NHSO and SSO has established some degree of technical capacities to enable appropriate service provision, monitor quality of care, protect consumer rights. We do not have much evidence to justify SSO capacities to monitor fraud and take legal actions. These indicate some room for improvement in terms of organizational capacities.

The second factor is the degree of provider autonomy. High degree of provider autonomy (e.g., in financial management, arrangement of service provision) stimulates innovation and efficiency of resource use but the main drawback is that it also allows potential opportunistic behaviours [4]. For both UCS and SHI, capitation method and relative weight DRG allow both public and private providers to retain surplus and flexibilities in internal management of allocated resource. In this study, some profit-making activities such as cherry picking and frauds as well as some innovations such as on-site service provision were reported. However, a notable innovation from provider is not seen in this study.

Third, our findings concurred with the literature about power balance between purchasers and providers. International experience indicated that power imbalance hinders provider's performance and willingness to improve. Dominant providers, in our case, high-end private

hospitals, ignored the public purchaser's incentives as they sought for profit-making markets [37]. NHSO and SSO are national level and powerful purchasers, but they still have to rely on private contracting especially in urban setting. They do not allow providers to negotiate the payment rates as the rates are guided by cost evidence. However, the providers could negotiate over the larger size of registered members. Both purchasers also felt reluctant to introduce more criteria and rigorous measures to detect low quality and false claims because they needed more private providers to engage.

Prior studies show mixed evidence on merits of contracting private sector health services on access and equity in access to health services [38, 39]. For example, the South Korean contracting influenza vaccination to private providers neither improved vaccination coverage nor reduced inequity, similar to studies in Cambodia and Guatemala [40, 41]. The inconclusive outcomes of contracting can be influenced by market structure, contextual environment and purchasing design. There is little evidence on its impacts on quality and efficiency [39]. These are also our knowledge gap in Thailand.

On access, the 2019 national Health and Welfare Survey shows that an annual utilization rate among 20–60 years old SHI members was lower than that of UCS members (0.89 versus 1.24 visits per capita) [12]. e among SHI members was probably a result of more limited access to contracted tertiary care hospitals than compared to UCS members who accessed care from their registered primary care units. However, our study scope does not allow us to provide much information about various aspects of contracting outcomes; a more comprehensive analysis is required.

## 5. Conclusions

To fill the public sector healthcare facility gaps in urban settings, insurance funds inevitably have to enter into contractual agreements with existing private healthcare facilities. The private-for-profit contracted providers require the insurance fund's strong regulatory capacity, enforcement and auditing measures in parallel with financial incentives to achieve the goal of access to quality care and respond to national health goals such as NCD, prevention and health promotion. Though findings from this study are context specific, it contributes to a better understanding on contractual arrangements between private health facilities and public health financing institutions.

## Author Contributions

**Conceptualization:** Woranan Witthayapipopsakul, Waritta Wangbanjongkun.

**Data curation:** Aniqa Islam Marshall, Somtanuek Chotchoungchatchai.

**Formal analysis:** Aniqa Islam Marshall, Woranan Witthayapipopsakul.

**Investigation:** Aniqa Islam Marshall, Woranan Witthayapipopsakul, Somtanuek Chotchoungchatchai.

**Methodology:** Aniqa Islam Marshall, Woranan Witthayapipopsakul.

**Project administration:** Somtanuek Chotchoungchatchai, Waritta Wangbanjongkun.

**Supervision:** Viroj Tangcharoensathien.

**Writing – original draft:** Aniqa Islam Marshall, Woranan Witthayapipopsakul.

**Writing – review & editing:** Aniqa Islam Marshall, Woranan Witthayapipopsakul, Viroj Tangcharoensathien.

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
