## [Decision Letter · Decision Letter 0]

13 Sep 2022

PGPH-D-22-01043

Contracting the Private Health Sector in Thailand’s Universal Health Coverage

Dear Mrs. Aniqa Islam Marshall,

Thank you for submitting your manuscript to PLOS Global Public Health. After careful consideration, we feel that it has merit but does not fully meet PLOS Global Public Health’s publication criteria as it currently stands. Therefore, we invite you to submit a revised version of the manuscript that addresses the points raised during the review process.

We look forward to receiving your revised manuscript.

Kind regards,

Genevieve Cecilia Aryeetey, Ph.D

Academic Editor

Journal Requirements:

Additional Editor Comments (if provided):

Reviewers' comments:

Reviewer's Responses to Questions

**Comments to the Author**

1. Does this manuscript meet PLOS Global Public Health’s publication criteria? Is the manuscript technically sound, and do the data support the conclusions? The manuscript must describe methodologically and ethically rigorous research with conclusions that are appropriately drawn based on the data presented.

Reviewer #1: Partly

Reviewer #2: Yes

2. Has the statistical analysis been performed appropriately and rigorously?

Reviewer #1: N/A

Reviewer #2: N/A

3. Have the authors made all data underlying the findings in their manuscript fully available (please refer to the Data Availability Statement at the start of the manuscript PDF file)?

Reviewer #1: No

Reviewer #2: No

4. Is the manuscript presented in an intelligible fashion and written in standard English?

Reviewer #1: Yes

Reviewer #2: Yes

5. Review Comments to the Author

Reviewer #1: Major Corrections:

The study has some serious shortcomings which must be resolved.

1. The findings seems to be largely based on the document review, and the analysis of indepth interviews is insignificant. The authors must clearly specify the purpose of the indepth interviews and present its analysis and finding in a distinct manner.

2. The conclusions drawn from the study does not align with the analyses and findings. The study fails to establish how contracting with the private healthcare facilities has filled the gap between demand and supply of the "quality care". Authors must include required evidence in the analysis to support their conclusions.

3. The text is repetitive at many places. The authors must revise the manuscript to make it more succinct and to the point.

Minor Corrections:

Page-5 Line-6: There is typographical error the word "proving" make in sense in the statement.

Page-16 Line-8 The line should be rephrased to make it clearer.

Page-16 Line-11 & 12 are repetition of text in line 3& 4.

Page-16 Line-14-12 The paragraph gives contradicting information. Line 15 tells 100 bed as minimum resource and standard requirements for all type of facilities but the line 18 say 150 beds for private facilities.

Page-18 Section 3.2 It is informed that NHSO can contract all three categories but "In practice NHSO primarily contracts with Main Contracting Units". The reason for such an anomaly should be explained to give readers more insights into the challenges of contracting.

Page-19 Section 3.3 Data on the ratio of applications and selected contracting unit will be helpful for the reader to understand the openness of competition and process efficiency of contracting.

Page-22 Line-8 The word "Kind" might have different meanings in different contexts. It should be specified and made clear with the use of examples in the context of study.

Page-23 Line-10 The idea of "relative weight" used for payments may be hard to understand for global readers. It should be elaborated briefly.

Page-23 Line-18 The text should be rephrased to make it comprehensible.

Page-23 Line 19-21 The idea that Supra- and sub-contractors deal with main contracting agency for all the purposes and not SSO is in repetition. This repetition should be removed.

Page-25 Line 3-7 Grammatical errors in the sentence. Check the usage of verb "compliance".

Page-26 Line 1 The sentence should be rephrased to remove repetition of words and clarity.

Reviewer #2: This is a comprehensive description of two purchasers’ approaches to contracting out with private hospitals and clinics in Thailand within its universal health coverage system. Using descriptive case study methods, the study provides detailed insights into the

Major comments.

1. The authors could better situate this study within the broader literature, specifically on strategic purchasing – and the way in which contracts are a component of strategic purchasing to achieve health system objectives.

Some key papers seem to be missing, e.g., this paper from Thailand, https://www.ncbi.nlm.nih.gov/pmc/articles/PMC6706967/#R9

And this paper from SSA may also be worth reviewing - https://www.tandfonline.com/doi/full/10.1080/23288604.2022.2051795

If the authors could provide a summary of this broader literature on strategic purchasing in the introduction, and then again in the discussion, this would help to interpret the findings, and the limitations and challenges with the contracting models in Thailand.

2. It would also help to introduce the discussion of fraud earlier on in the study. Could the authors clarify why the case study methods didn’t include a description of the incentives or mitigation strategies against fraudulent behavior or other aspects of poor performance.

3. The authors appears to be biased in favor of contracting private providers, and perhaps they could provide more nuanced presentation of the strengths and challenges drawing on the theoretical and empirical literature. E.g., this statement is overly strong and the empirical basis is limited – “contracting private providers are effective and should be advocated in developing countries.”

6. PLOS authors have the option to publish the peer review history of their article (what does this mean?). If published, this will include your full peer review and any attached files.

**Do you want your identity to be public for this peer review?** For information about this choice, including consent withdrawal, please see our Privacy Policy.

Reviewer #1: No

Reviewer #2: No

---

## [Decision Letter · Decision Letter 1]

3 Apr 2023

Contracting the Private Health Sector in Thailand’s Universal Health Coverage

PGPH-D-22-01043R1

Dear Mrs. Marshall,

We are pleased to inform you that your manuscript 'Contracting the Private Health Sector in Thailand’s Universal Health Coverage' has been provisionally accepted for publication in PLOS Global Public Health.

Best regards,

Sanjana Ravi, PhD, MPH

Academic Editor

Reviewer Comments (if any, and for reference):

Reviewer's Responses to Questions

**Comments to the Author**

1. If the authors have adequately addressed your comments raised in a previous round of review and you feel that this manuscript is now acceptable for publication, you may indicate that here to bypass the “Comments to the Author” section, enter your conflict of interest statement in the “Confidential to Editor” section, and submit your "Accept" recommendation.

Reviewer #1: All comments have been addressed

Reviewer #2: All comments have been addressed

2. Does this manuscript meet PLOS Global Public Health’s publication criteria? Is the manuscript technically sound, and do the data support the conclusions? The manuscript must describe methodologically and ethically rigorous research with conclusions that are appropriately drawn based on the data presented.

Reviewer #1: Yes

Reviewer #2: Yes

3. Has the statistical analysis been performed appropriately and rigorously?

Reviewer #1: Yes

Reviewer #2: N/A

4. Have the authors made all data underlying the findings in their manuscript fully available (please refer to the Data Availability Statement at the start of the manuscript PDF file)?

Reviewer #1: Yes

Reviewer #2: Yes

5. Is the manuscript presented in an intelligible fashion and written in standard English?

Reviewer #1: Yes

Reviewer #2: Yes

6. Review Comments to the Author

Reviewer #1: None

Reviewer #2: This is an improved paper that addresses all of my comments provided earlier.

7. PLOS authors have the option to publish the peer review history of their article (what does this mean?). If published, this will include your full peer review and any attached files.

**Do you want your identity to be public for this peer review?** For information about this choice, including consent withdrawal, please see our Privacy Policy.

Reviewer #1: **Yes: **Dr. Rakesh Chandra

Reviewer #2: No
